# Strategies for Enhancing the Multi-Stage Classification Performances of HER2 Breast Cancer from Hematoxylin and Eosin Images

**DOI:** 10.3390/diagnostics12112825

**Published:** 2022-11-16

**Authors:** Md. Sakib Hossain Shovon, Md. Jahidul Islam, Mohammed Nawshar Ali Khan Nabil, Md. Mohimen Molla, Akinul Islam Jony, M. F. Mridha

**Affiliations:** Department of Computer Science, American International University-Bangladesh, Dhaka 1229, Bangladesh

**Keywords:** breast cancer, HER2, modified Xception, Grad-CAM, StarDist, multi-stage, nuclei segmentation

## Abstract

Breast cancer is a significant health concern among women. Prompt diagnosis can diminish the mortality rate and direct patients to take steps for cancer treatment. Recently, deep learning has been employed to diagnose breast cancer in the context of digital pathology. To help in this area, a transfer learning-based model called ‘HE-HER2Net’ has been proposed to diagnose multiple stages of HER2 breast cancer (HER2-0, HER2-1+, HER2-2+, HER2-3+) on H&E (hematoxylin & eosin) images from the BCI dataset. HE-HER2Net is the modified version of the Xception model, which is additionally comprised of global average pooling, several batch normalization layers, dropout layers, and dense layers with a swish activation function. This proposed model exceeds all existing models in terms of accuracy (0.87), precision (0.88), recall (0.86), and AUC score (0.98) immensely. In addition, our proposed model has been explained through a class-discriminative localization technique using Grad-CAM to build trust and to make the model more transparent. Finally, nuclei segmentation has been performed through the StarDist method.

## 1. Introduction

Malignancies are responsible for a large number of fatalities. Some of the different types of cancer that exist today are bladder cancer, colorectal cancer, thyroid cancer, breast cancer, etc. Most of these malignancies affect women, with breast cancer being one of the most prevalent. In 2020, 12% of malignant tumors in the human population were caused by breast cancer [1]. By 2040, the number of cases is predicted to increase by more than 46% [2]. Breast cancer remains the second most lethal cancer diagnosis, even though mortality rates from the disease fell by 1% in 2013, possibly due to therapeutic advancements.

The most lethal and often diagnosed breast cancer in women is HER2, one of the several subtypes of breast cancer. Trastuzumab, a form of HER2-targeted medicine, was recently introduced, and as a result, patients with HER2-positive breast cancer now have far better prospects for survival [3]. About twenty percent of women still develop metastases despite taking trastuzumab and adjuvant chemotherapy for their HER2-positive breast cancer [4]. However, an early breast malignancy diagnosis can improve the chances of survival. Hematoxylin and Eosin (H&E) are standard techniques used by pathologists to determine morphological information, including the shape, pattern, and structure of the cells and tissues that aid in diagnosing cancer. A further staining method called immunohistochemistry (IHC) is used to verify breast cancer. Moreover, the IHC staining method using antibodies highlights several antigens, including HER2, progesterone receptor (PR), and estrogen receptor (ER) [5]. The result of IHC staining can be divided into positivity scores between 0 and 3+. A positivity score of 0 and 1+ is defined as HER2-negative (HER2-). On the contrary, a score of 3+ is considered HER2-positive (HER2+). However, a score of 2+ requires further testing using ISH to determine HER2 gene status [6]. Medicines such as trastuzumab exist for treatment but are expensive and have harmful side effects [7].

An essential step in determining the type of lesion for initial diagnosis is the hematoxylin and eosin (H&E) stain study of breast tissue biopsy. Hematoxylin is responsible for the purple stain of the nuclei, and the pinkish hue is mainly due to cytoplasm. The grade of the carcinoma can be determined by the pathologist using this staining, which helps to explore the patient’s treatment options. Normal tissue, benevolent lesions, in situ carcinomas, and invasive carcinomas are the four categories into which H&E stain images can be divided. In H&E-stained slides, normal breast tissues display substantial amounts of cytoplasm (pinkish patches) and densely packed nuclei forming glands [8]. A benign lesion is made up of numerous nearby clusters of tiny nuclei. Benign lesions can progress, if left untreated, into situ carcinoma, which seems surrounded by circular clusters while losing some of its glandular characteristics. The bigger nuclei in invasive carcinoma lose their clustered structure and fragments to the surrounding regions.

Computer-Aided Diagnosis (CAD) systems include image exploration and machine-learning techniques created to aid doctors in diagnosis. Their use can improve diagnostic accuracy while expediting the diagnostic procedure [9,10]. Computer-assisted image analysis systems have been created to help human pathologists achieve precise results. To limit costs and lower the risk of death, new and better deep-learning approaches are being developed to identify breast malignant (HER2) cells in their preliminary stages.

A large dataset is also necessary for training a DL model, which extends the training time. While using a new, small dataset for training to conduct research, a method known as Transfer Learning (TL) can shorten training time and enhance model performance. Any DL model, such as CNN, can be utilized to perform TL using one of three methods. First, a feature extractor might be a pre-trained CNN model. The second technique entails adjusting the final layer weights of a pre-trained CNN, while the third technique involves doing the same for the architecture as a whole [5].

In our study, we have presented a modified TL architecture called ‘HE-HER2Net’ based on Xception from H&E images of the BCI dataset. Compared to all existing models, this proposed model is robust enough to obtain worthy performance in Accuracy, Precision, and Recall. In addition, several optimization techniques have been integrated into our proposed model to abate underfitting and overfitting problems, reduce the complexity of the network, extract more feature information, etc. Moreover, layer-wise explanation has been visualized to explain the model output intuitively. Finally, by analyzing instances of H&E images, our work has accomplished a star-convex-based method. The overall contribution of the report is given briefly.

Introducing effectiveness of Additional Global Average Pooling Layer, Dropout layers, Batch Normalization layers, Dense layers with a Swish activation function, and Classifier layer with SoftMax activation layer.Comparing the proposed model to other existing models such as VGG19, NASNetLarge, EfficientNetB7, Inception V3, ResNetV2, InceptionResNetV2, DenseNet201, and Xception.Comparing the proposed model’s performances with several optimizers, such as Adam and Adagrad, and activation functions, such as ReLU and Swish.Explaining our model through Grad-CAM to explain how the model works.Additionally, segmenting nuclei of the H&E images through the StarDist method.

The paper is organized as follows. In Section 2, we discuss the related work. In Section 3, we explain the materials and methods. Section 4 and Section 5 describe the results and discussion, and limitations, respectively. Finally, Section 6 includes the conclusions of our research.

## 2. Related Work

Traditional machine learning (ML) and deep learning (DL) are the two computational techniques for pathological images. ML algorithms, which are frequently utilized in the field of prognostic prediction, can significantly minimize the amount of time that the diagnostic procedure takes. Expert pathologists use pricey microscopes and manual procedures to identify HER2 and its state from H&E and IHC stains [11]. However, they involve human interpretation, and such HER2 status detection techniques are liable to inaccuracy [12]. As a result, scientists worldwide have created a range of automated techniques for classifying HER2 status from IHC and H&E-stains. MRI and ultrasound images were also used in a study to classify HER2 status [13,14]. A Support Vector Machine (SVM) was used there as the approach to identify the HER2 status in MRI images.

DL has made significant strides in recent years, benefiting numerous industries, including health. CNN is a DL network type; in particular, it has been demonstrated to be effective in several classification tasks. CNN identifies histopathological abnormalities in regular H&E pictures that are associated with the presence of atomic biomarkers in a range of cancer types, which include colorectal [15], lung [16], prostate [17], and skin cancers [18]. Moreover, researchers can use CNN for cell segmentation or detection, tumor classification, and carcinoma localization in digital pathology.

To reliably diagnose illnesses, it is crucial in clinical practice to correctly categorize histopathological pictures. This type of operation may be automated with DL, particularly TL, to replace the time-consuming and expensive labor effort of human specialists and satisfy the requirements for high accuracy, extended data sizes, and efficient computing. TL is frequently employed because there are not many huge, publicly accessible, annotated digital slides. TL addresses the problem of cross-domain learning by transmitting relevant knowledge from the source domain to the task domain [19]. Deep TL is frequently used because of its improved performance and adaptability [20,21,22,23,24].

Oliveira et al. [25] developed a CNN model based on multiple instance learning (MIL) approaches identifying HER2 status from H&E images. Initially, the CNN model was pre-trained from IHC images on the HER2SC dataset. Finally, the author trained their model with H&E images from the HER2SC dataset and tested H&E stained slides from the CIA-TCGA-BRCA (BRCA) dataset. As a result, they obtained test accuracies of 83.3% and 53.8%, respectively, from these datasets.

H&E stain images were used in [7,26] to determine the HER2 status. U-Net was utilized in the framework in [27] to find nuclei locations in the WSI regions of the H&E-stain. To categorize HER2- or HER2+, it also used a cascade of CNN architecture. The proposed methodology obtained an AUC value of 0.82 in the Warwick dataset [28] and 0.76 AUC in the TCGA-BRCA dataset. However, the suggested technique needs to report patch-level and slide-level AUC independently. Furthermore, the method struggled to locate HER2+ cells with a score of 2+ (0.73 AUC). To predict the DAB density from H&E-stained WSIs, and HER2 scores from produced DAB density, W. Lu developed a GNN-based system [27]. In the TCGA-BRCA test set, the architecture, as mentioned earlier, achieved an AUC of 0.75, whereas the HER2C and Nott-HER2 datasets yielded AUCs of 0.78 and 0.80. However, while testing the model, a HER2 score of 2+ was avoided.

Shamai et al. [29] have tried to forecast the expression of molecular biomarkers in breast cancer simply using the analysis of digitalized H&E-stained tissues. To predict biomarkers, including ER, PR, and HER2, from tissue morphology, a deep CNN based on residual network (ResNet [30]) architecture was created in this study. The AUCs for these three biomarkers were 0.80, 0.75, and 0.74, respectively. Two significant limitations were that the data came from a single organization (Vancouver General Hospital) and only contained TMA pictures from 5356 patients, not WSI. Furthermore, Naik et al. [31] created multiple instances of DL-based neural networks to predict the same molecular indicators from H&E-stained WSI.

From the above discussion, most existing studies have been conducted dealing with predicting different subtypes from histopathological images, defining binary expression labels, generating images, etc. However, unfortunately, there has not been any proper research on HER2 breast carcinoma from H&E images dealing with four-class multi-stage classification problems on the BCI dataset. Hence, this inspires us to conduct this multi-stage classification problem of HER2 breast cancer. In our work, we have proposed a modified TL-based model to solve this multiclass problem on the BCI dataset.

## 3. Materials and Methods

### 3.1. General Overview of the Method

To first obtain a multi-stage classification of HER2 breast cancer from H&E (hematoxylin and eosin) images on the BCI dataset, we trained several ImageNet weight-based transfer learning models, such as Vgg19, NASNetLarge, EfficientNetB7, InceptionV3, InceptionResNetV2, ResNet152V2, DenseNet201, and Xception. This base model was not robust enough to efficiently perform multi-stage classification because of partial data from the H&E dataset. Hence, these base models obtained extremely low accuracy, precision, recall, and AUC value for this problem. In addition, the loss value was unexpectedly much higher than the considerable value. Therefore, it is essential to reduce the loss value when applying this model to classification problems. We used different modified models suitable for this dataset to obtain a higher performance score and minimize loss. Among these base models, InceptionResNetV2 and Xception performed better, bringing the same accuracy, precision, and recall AUC score compared to all other existing models. In addition, the loss value was significantly lower in these models. As this model was much more reliable and robust according to the performance, we further modified this model by replacing the flattened layer of the base model with global average pooling. In addition, we added several dropout layers and dense layers with different activation functions (ReLU, swish), applying a batch normalization layer to the base model. Thus, we attained the best-modified model that acquired significantly better results. For all the implementation processes, we used early stopping by setting up monitor = “val_loss”, mode = “min”, patience = 3, restore_best_weights = True; to overcome the overfitting problem. We set the Adam optimizer to have a learning rate of 1 × 10^−5^, the loss function to categorical_crossentrophy, and the metrics function to obtain accuracy, precision, recall, and AUC values for all the CNN (Convolution Neural Network) models. For the modified model, we explored different activation functions, optimizers with different learning rates, and regularizes for hyperparameter tuning. In addition, we applied data augmentation to the modified model to get better performance after applying several combinations. Our proposed model, ‘HE-HER2Net, ’ outperformed all performance evaluation metrics compared to all base models, including other modified combinations of the improved versions.

Later, we explained our proposed model through Grad-CAM by generating a heatmap of the convolution layer of HE-HER2Net to analyze our model’s robustness and weakness and for the decision-making to observe intuitively. Additionally, we performed nuclei segmentation using StarDist, randomly taking four images from the distinct stages of the BCI-H&E image to visualize the nuclei of the H&E image. StarDist achieved satisfying results as the nuclei of the H&E images are roundish in shape.

### 3.2. Dataset Description

A new breast cancer immunohistochemical (BCI) benchmark dataset [32] has been applied in this research. Initially, Hamamatsu NanoZommer S60 was used as the data scanning ingredient, where the scanning resolution was 0.46 µm per pixel. About 600 WSI slides have been scanned. Each of the slides contains 20,000 pixels. Later on, each of the slides was divided into 16 blocks with a resolution of 1024 ∗ 1024. This BCI dataset contains 4870 pairs of H&E & IHC images of 1024 ∗ 1024 resolution and includes four categories of 0, 1+, 2+, and 3+, as illustrated in Figure 1.

To perform multi-stage classification from hematoxylin and eosin-stained images, we have taken only H&E images, where 3896 images have been set for the training dataset and 977 images for the test dataset. There are other publicly available histopathological datasets. However, as far as we know, no suitable histopathological dataset contains various categories according to the direction of CAP/ASCO [6] to classify multiple stages of breast cancer. Hence, we have used this dataset in our research to attain our goal.

### 3.3. Environment Setup

We have trained all the pre-trained models using Keras and TensorFlow libraries in google Colab by taking different suitable input sizes, batch sizes, number of epochs, augmentation parameters, optimizers with various learning rates, activation functions, etc., and we resized all our models by defining the proper input shape according to the model requirements. It is an unavoidable step to resize all the images into a fixed size. We also shortened the original pixel value of 1024 ∗ 1024 to a lower pixel to efficiently train and accelerate the training time. For the base model, we did not apply any data augmentation. We kept the same optimizer (Adam), batch size (16), learning rate (1 × 10^−5^), activation function (ReLU), epochs (50) with early stopping, and performance metrics setting up accuracy, precision, recall, and AUC. In addition, for the all-modified models, we applied decent augmentation (width_shift_range = 0.2, height_shift_range = 0.2, rotation_range = 0.2, vertical_flip = True), different optimizers (ReLU, Swish), different learning rates (1 × 10^−3^, 1 × 10^−5^), and epochs (80), to apply hyperparameter tuning in our modified proposed method to achieve the best result. The summary of the environmental setup of this study is highlighted below in Table 1.

### 3.4. Proposed Architecture HE-HER2Net

For the multi-stage classification of the histopathological images from the BCI dataset, we proposed a transfer learning method based on Xception, known as HE-HER2Net.

Having a lack of abundance of data transfer learning methods can save not only training time but also computation costs. In this study, we modified a robust pre-trained method known as Xception [33], which has been trained on the ImageNet dataset. Xception consists of 36 convolution layers, forming the feature education base of the model. It refers to an extreme version of the Inception model with a modified depth-wise separate convolution that performs better than Inception. In this network, data goes to the entrance flow, then through the central flow, repeating eight times, and finally, data passes through the external flow. All convolution and separable convolution layers are followed by batch normalization. Figure 2, given above, narrates our proposed workflow. In addition, Table 2 demonstrates the parameters of the additional layers of HE-HER2Net.

As we were working on the BCI-H&E dataset, which contains some misclassified data on the different stages of HER2 breast cancer, we applied several strategies to mitigate this bias problem, obtaining satisfying results. At first, we resized the H&E images to 299 ∗ 299 pixels, rescaled all images, and used data augmentation to get better prediction accuracy and overcome the overfitting problem. Next, we removed the default classification layer to perform four class problems. Then, we introduced global average pooling, replacing the flattened layer as it is more natal to the convolution structure because it drives correspondence between feature maps and categories. Moreover, it lessens the overfitting problem by decreasing the total number of parameters in the model. Finally, we experimented with diverse regularization techniques to improve model performance. We applied dropout regularization (0.3) [34] before each dense layer to prevent overfitting. Dropout drops randomly selected neurons during training, which helps the model accuracy gradually increase and decrease loss. Figure 3 illustrates how dropout works.

As described in the proposed model, we added a dense layer with a Swish activation function after each dropout layer and before the batch normalization layers. Using these dense layers, maintaining the proper way classifies more features provided by convolution layers. In our model, the dense layer empowered our network’s ability to organize better-extracted elements. We experimented by taking ReLU and Swish activation functions in our model. Swish outperformed the result of using ReLU in every performance metric. The claim experimented by Prajit et al. [35] showed the Swish activation function performed better than ReLU on complex datasets. This smooth, non-monotonic function converges quicker and allows data normalization. This activation function is defined below.
(1)f(x)=x−σ(x)
where σ(x)=(1+exp(−x))−1 is the sigmoid activation function.

In addition, we employed batch normalization [36] layers between dense and dropout layers by normalizing the hidden layer activation. It speeds up the training process, solving the internal covariate shift problems to ensure every input for every layer is distributed around the same mean and standard deviation. The mathematics behind the batch normalization is specified as follows. Here, $x_{i}$ = inputs over a minibatch size m, $\mu_{B}$ = means and $\sigma_{B}^{2}$ = variance.
(2)$$\;\mu_{B}=\frac{1}{m}\sum_{i=1}^{m}x_{i}\;$$
(3)$$\;\sigma_{B}^{2}=\frac{1}{m}\sum_{i=1}^{m}{\left(x_{i}-\mu_{B}\right)}^{2}\;$$
Now the samples with zero means and unit variance are normalized. Here, ϵ is used for numerical stability, avoiding zero in the denominator and $\hat{x_{i}}$ = activation vector.
(4)$$\;{\hat{x}}_{i}=\frac{x_{i}-\mu_{B}}{\sqrt{\sigma_{B}^{2}+\epsilon }}\;$$
Finally, we get the following equation after the scaling and shifting process. Here, $y_{i}$ = output.
(5)$$\;y_{i}=\gamma {\hat{x}}_{i}+\beta \;$$

Here, γ and β are learnable parameters. Finally, as a classification layer of four classes (HER2-0, HER2-1+, HER2-2+, HER2-3+), we employed a dense classification layer of four neurons along with a SoftMax [37] activation function. The SoftMax function is widely used as a multiclass classification problem, as it returns the probability of each class, ranging between 0 and 1. Using SoftMax, the target class gets a high probability. The SoftMax activation function is described below.
(6)$$\;\mathrm{softmax}\left(z_{i}\right)=\frac{\exp\left(z_{i}\right)}{\sum_{j}\exp\left(z_{j}\right)}\;$$
Here, *z* = values from the neurons of the output layer, and exp acts as the nonlinear function.

We experimented with taking optimizers (Adam, Adagrad) with different learning rates for hyperparameter tuning. In our proposed model, we used the Adam optimizer with a learning rate of 1 × 10^−5^. As we performed a multiclass classification problem, we used catagorical_crossentrophy as the loss function. The mathematical explanations of categorical_crossentrophy are described as follows.
(7)$$\;L_{i}=-\sum_{j}t_{i,j}\log\left(p_{i,j}\right)\;$$
where

*p* = predictions

*t* = targets

*i* = data points and

*j* = class

This loss function is for multi-category classification problems and SoftMax output units.

To evaluate our model performance, we calculated accuracy, precision, recall, and AUC as the performance metrics. As a result, our proposed model, ‘HE-HER2Net’, outperformed every existing model with significant changes in accuracy, precision, recall, and AUC. In addition, our model reduced the loss value more than all other models. Finally, we explain our model through Grad-CAM.

### 3.5. Model Explainability Using Grad-CAM

In terms of building trust models in intelligent systems based on CNN networks, it is essential to clarify how these models are predicting and what is being predicted. To establish adequate trust and confidence, we explained our model through a visual explanation using Gradient-Weighted Class Activation Mapping (Grad-CAM) [38], which is known as a class-preferential localization technique that generates graphic descriptions of a model. It uses the gradient instructions flowing into the last convolutional layer to assign significant values to each neuron.

To get the localization map, at first, Grad-CAM computes gradient $y^{c}$ concerning feature map A of a convolutional layer. After that, these gradients are global average-pooled to attain neuron weights. Finally, the heatmap is generated by performing a weighted combination of feature maps followed by a ReLU. The mathematical intuition behind Grad-CAM is given below.
(8)$$\;\alpha_{k}^{c}=\overset{\mathrm{global}\;\mathrm{average}\;\mathrm{pooling}\;}{\overset{︷}{\frac{1}{Z}\sum_{i}\sum_{j}}}\underset{\mathrm{gradients}\;\mathrm{via}\;\mathrm{backprop}\;}{\underset{︸}{\frac{\partial y^{c}}{\partial A_{ij}^{k}}}}\;$$
(9)$$\;L_{\mathrm{Grad}-\mathrm{CAM}\;}^{c}=\mathrm{ReLU}\underset{\mathrm{linear}\;\mathrm{combination}\;}{\underset{︸}{\left(\sum_{k}\alpha_{k}^{c}A^{k}\right)}}\;$$
Here,

$y^{c}$ = Score of class c of a network before SoftMax.

$A^{k}$ = Feature map activations.

$\alpha_{k}^{c}$ = Neuron weights.

*Z* = Number of pixels in the feature map.

In our study, we took several convolution layers (block1_conv1, block5_sepconv1_act, block10_sepconv1_act, block14_sepconv2_act) in different stages of the model and analyzed our model for multiple classes of H&E images through Grad-CAM. Here, in the first convolution layer (block1_conv1), we see all the contours and borders of the images have been pointed. By looking at the convolution layers (block5_sepconv1_act, block10_sepconv1_act), it is clear that the layers are trying to detect concepts in the image. According to the author of Grad-CAM, it can be assumed that the last convolution layer has the best spatial information. Hence, we analyzed the last convolution layer (block14_sepconv2_act) to obtain how our model was performing classification based on the potential part of the image. From the output of the generated heatmap of different stages of the H&E image, we see various parts of the images have been highlighted. This indicates that our model classified multiple stages of H&E images, focusing on these highlighted areas of the image.

### 3.6. Nuclei Segmentation Using StarDist

Nuclei segmentation from histopathological images is very important for helping pathologists and researchers analyze whether cells are benign or malignant. Generally, cancer cell nuclei are more extensive and darker compared to normal cells because they contain comprehensive DNA. Thus, nuclei segmentation is an essential task for researchers in digital pathology so that researchers can perform a lot of quantitative analysis by analyzing shape, texture, size, etc. In addition, nuclei segmentation from histopathological images can be obtained as the input of the CNN classifier since the nuclei are the most important instances in histopathological images. Some research has been conducted for nuclei segmentation using the state of methods, such as Mask R-CNN and U-Net. However, these models could not give satisfactory results. Uwe Schmidt et al. [39] performed an experiment applying Mask R-CNN on the TOY dataset and showed worse outcomes due to many overlapping bounding boxes and touching pairs of objects. They also performed another experiment on the TRAGEN dataset using U-Net. This model also showed a low-performance evaluation score due to the abundance of touching cells. To mitigate these problems of crowded cells, the author proposed nuclei cell localization via star-convex polygons, which outperformed the existing state-of-the-art model.

Solving all these limitations, we used the StarDist method for nuclei segmentation so that, despite having crowded cells, nuclei of the H&E images can be segmented precisely compared to other models. A prominent issue for this model is that objects must be star-convex illustrated in Figure 4. It means that the center point of an object must be connected in a straight manner to all boundary points. Otherwise, the object will not be detected by the model. We have taken one random image from each of the stages of the H&E dataset, then segmented it using the StarDist method to visualize and leverage the ability of the StarDist process. In addition, we took the ‘2d_versatile_he’ pre-trained model from the StarDist method, as our research was based on the H&E image. It is important for this model that objects must be star-convex, which means the center point of an object must be connected straightly to all boundary points.

### 3.7. Evaluation Metrics

To evaluate the performance of our proposed model, we obtained different evaluation metrics such as accuracy (ACC), precision (P), recall (R), and AUC. In addition, we computed a confusion matrix, which is not exactly a performance metric, but based on this, other performance metrics are calculated. The confusion matrix visualizes the ground truth labels vs. predicted labels. Each row of the confusion matrix defines instances in a predicted class, and each column describes instances in an actual class. Terms such as True Positive (TP), True Negative (TN), False Positive (FP), and False Negative (FN) depend on the confusion matrix.

Here,

TP = Model correctly predicted a number of positive class samples.

TN = Model correctly predicted a number of negative class samples.

FP = Model incorrectly predicted a number of negative class samples.

FN = Model incorrectly predicted a number of positive class samples.

The mathematical intuition of accuracy (ACC), precision (P), and recall (R) are described as follows.
(10)$$\;\mathrm{ACC}=\frac{\mathrm{TP}+\mathrm{TN}}{\mathrm{TP}+\mathrm{TN}+\mathrm{FP}+\mathrm{FN}}\;$$
(11)$$\;P=\frac{\mathrm{TP}}{\mathrm{TP}+\mathrm{FP}}\;$$
(12)$$\;R=\frac{\mathrm{TP}}{\mathrm{TP}+\mathrm{FN}}\;$$

Here,

0 < P < 1 and 0 < R < 1.

Moreover, we measured the AUC value, which refers to the area under the ROC curve. The AUC defines a classifier’s performance. It indicates how well the model differentiates between the given instances. The AUC ranges from 0 to 1. The higher the AUC, the better the model’s prediction.

## 4. Results and Discussion

In this experiment, we proposed ‘HE-HER2Net’ based on the transfer learning method to classify multi-stage classifications of HER2 breast cancer. We compared our model with the existing CNN model to leverage our model’s robustness. Several performance metrics, such as accuracy, precision, recall, and AUC value, were computed to analyze our model. The best performance we achieved from the base model was Xception and InceptionResNetV2. However, their performance was not close to the considerable minimum level. To alleviate this problem further, we modified the Xception and InceptionResNetV2 models to get gratified results. It is important to classify whenever we are dealing with a lethal problem, such as cancer. After experimenting with several modifications, our proposed model extensively surpassed all existing models with a test accuracy of 0.87, a precision of 0.88, a recall of 0.86, and an AUC of 0.98, followed by the best base model with a test accuracy of 0.71, the precision of 0.73, recall of 0.69 and AUC of 0.90. We explored our proposed model with different optimizers (Adagrad), and with different activation functions ReLU. Our model, ‘HE-HER2Net’, performed better with the Swish activation function than with the Adam optimizer. Figure 5 demonstrates a comparative study of all the experimented models.

From the given illustration above, it can be clearly stated that HE-HER2Net surpassed all evaluation metrics for this classification problem. We obtained a confusion matrix to explore all the models deeply. Figure 6 describes all the confusion matrices.

Analyzing the confusion matrix, we see some base models, such as NASNetLarge, EfficientNetB7, ResNet152V2, and Vgg19, performed worse, whereas Xception and InceptionResNetV2 performed better, followed by InceptionV3 and DenseNet201. On the contrary, all the modified models obtained promising results. HE-HER2Net achieved the best performance overall. The diagonal deep blue color of the confusion matrix diagram represents how many instances the model predicted correctly compared with the ground truth value. To explain more of our model, the accuracy, loss, precision, recall, and AUC outputs are shown in Figure 7, Figure 8 and Figure 9.

By visualizing the graphs shown above, our model performed exceptionally well. Though validation curves of accuracy, precision, and recall were higher than training accuracy, precision, and recall, our model initially faced a small underfitting problem. Later, the ratio between the training and validation curve started to decrease, which means that our model improved and learned well from the training data. Moreover, when validation loss starts growing, the model begins causing an overfitting problem. Several actions have been taken in our work to eliminate the underfitting and overfitting problems. First, we set early stopping by monitoring validation loss for three consecutive epochs by setting up patient three, which is why none of our models have an overfitting problem. From the loss graph, we see that the loss value for the training data was always slightly higher than the validation loss value, which indicates our model performed well without having any underfitting problems. By observing the precision and recall graphs, we see that the training precision and recall values were not higher than the testing precision and recall, indicating no underfitting problem in our model. Moreover, the AUC graph shown in Figure 9 was also obtained to observe our model. In addition, Table 3 describes the results of our model compared with other models.

Visualizing the AUC graph demonstrates that our model obtained a satisfying AUC score of close to 1. Getting a higher AUC value indicates the robustness of classifying ability among several classes. Therefore, our proposed model performed the classification task amazingly well.

By observing the performance evaluation table, our model surpassed all other existing models with an accuracy of 0.87, a precision of 0.88, a recall of 0.86, and an AUC of 0.98. Among the base models, Xception and InceptionResNetV2 obtained almost similar results, but the results were not acceptable for this classification problem of breast cancer. As the used dataset, known as BCI, was biased and insufficient for training DL models, all the base models struggled to perform well. Introducing global average pooling, batch normalization, a dense layer with a Swish activation function, and a dropout mechanism significantly reduced these problems. Our proposed model works amazingly well for this classification problem from H&E images. Moreover, by comparing different optimizers and activation functions, we found that the Adam optimizer and Swish functions performed extremely well for this dataset.

We explained our model using Grad-CAM by analyzing different convolutional layers. Grad-CAM shows how models classify based on a particular area. It helps to make decisions by visualizing the import area of the region of the data image by generating a heatmap. Different steps of the model are described in Figure 10 to gain insight into how our model works step by step.

As the instances of a histopathological cell are very tiny and dense to visualize, it is tough to imagine the potential regions of the image. However, our model highlighted some specific areas generating bright heatmaps. We also explored the segmentation of the nuclei using the StarDist method. A visualization of the nuclei segmentation of the different classes is given in Figure 11, Figure 12, Figure 13 and Figure 14.

Nuclei segmentation based on StarDist worked well because of its star-convex shape. Therefore, any roundish shape objects can be predicted by applying StarDist.

Overall, HE-HER2Net obtained significant improvements. Moreover, more investigations are needed to solve the multiclass classification problems to diagnose the various stages of HER2 breast cancer. For nuclei segmentation, other state-of-the-art methods can be applied to compare robustness among different models for proper intuition.

## 5. Limitations

This study represents a modified robust model based on a pre-trained CNN for the multi-stage classification of HER2 breast cancer from H&E images. We have taken the BCI dataset only since no publicly available dataset contains four stages of HER2 breast cancer images. About 3896 H&E images have been used for the training set, and 977 H&E images have been utilized for the test set. As we know, DL models work well with a sufficient number of images. Unfortunately, we could not employ an adequate number of images in our model. Moreover, there have some bias problems in this dataset because it is very hard, even for a human, to differentiate images of different classes visually. That is why most of the base models in this study failed to perform minimum acceptable performance. All of the existing state-of-art models obtained accuracy, precision, and recall values within the range of 40–70%. As we know, these values are keenly connected with a confusion matrix; model robustness can also be explained from the confusion matrix, where each column and row represents an actual label and a predicted label, respectively. Therefore, analyzing each row and column, it is obvious how most of the base models struggled to accurately predict the actual label. During training, most of the base models faced underfitting and overfitting problems. In addition, none of the base models was robust enough to obtain spatial information precisely for this dataset. To accurately diagnose a lethal type of disease, such as breast cancer, it is necessary to build a robust model to handle biased datasets, obtaining a minimum loss value and optimizing overfitting and underfitting issues. That is why introducing global average pooling, more dense layers with a Swish activation function, batch normalization, and dropout layers to the base model alleviated these issues and significantly improved our proposed model. This study has some other drawbacks, for example, explaining the model with grad-CAM and segmenting nuclei cells with the StarDist method. Grad-CAM is a class-preferential localization technique that works intuitively for datasets where different classes within an image can be differentiated easily. In this study, in every image of H&E, several components within the image are hard to distinguish, for example, nuclei. That is why Grad-CAM visualizes the overall image by generating a heatmap. For nuclei segmentation, one criterion was that the objects in the image should be roundish. Therefore, StarDist works better when the nuclei cell is roundish; otherwise, it fails to detect nuclei properly. However, maintaining all of these issues, we tried our best to perform several tasks in this research.

## 6. Conclusions

Breast cancer is a very lethal and dangerous disease among women. Early diagnosis of HER2 breast cancer can help patients make decisions and start treatment with the help of Deep Learning. In this research, we investigated a transfer learning-based model to solve the multi-stage classification problem of HER2 breast cancer from hematoxylin and eosin images. First, we enquired about our research on the BCI (Breast Cancer Immunohistochemical) dataset, which comprises four types of HER2 images. However, this dataset is very complex and has a bias problem because it has similar kinds of data in each class. We experimented with several pre-trained models to achieve the best performance. However, all of the models showed unsatisfactory results. Most models had difficulty maintaining underfitting and overfitting issues, and acquiring acceptable accuracy. However, after several experiments, we proposed HE-HER2Net by introducing global average pooling, batch normalization layers, dropout layers, and dense layers with a Swish activation function to the base model of Xception. These additional blocks were robust enough to extract more pieces of information, train the model much faster and avoid overfitting issues during training. This proposed model, HE-HER2Net, surpassed all existing models in terms of accuracy, precision, recall, and AUC score. Next, we explained our model through Grad-CAM to make our model more transparent. Grad-CAM explains how our model learned for each of the convolution layers. Finally, we applied the StarDist method for nuclei segmentation, which precisely visualized all nuclei cells of the H&E images. Both pathologists and patients can benefit without costing much money and time for the diagnosis if breast cancer.

In conclusion, our future work will be on nuclei segmentation and color separation of breast cancer of histopathological images. In addition, another explainable model can be investigated and compared to make the model more transparent. Finally, more rigorous studies are needed to diagnose breast cancer so patients can reduce their risk and make proper decisions.

## Figures and Tables

**Figure 1 diagnostics-12-02825-f001:**
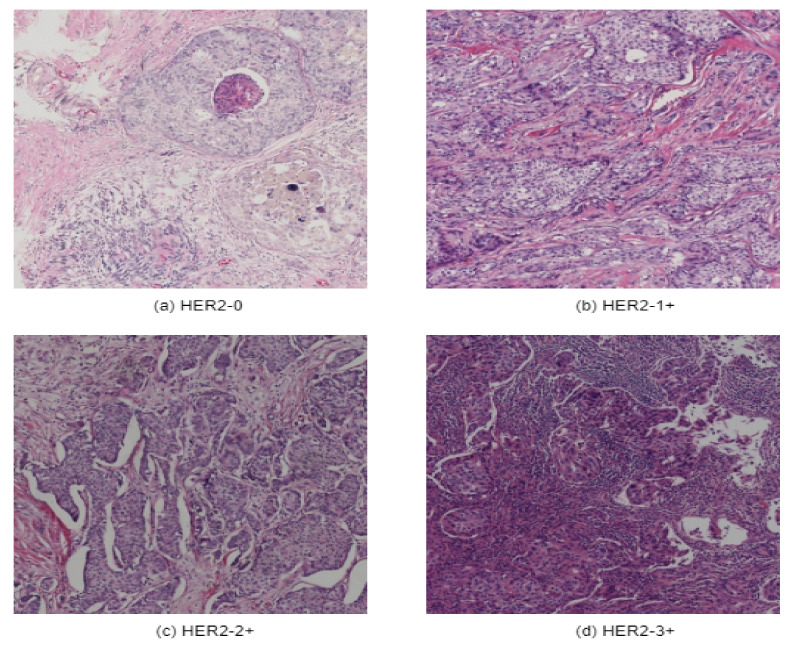
Sample images of each of the class of BCI-H&E dataset: (**a**) HER2-0, (**b**) HER2-1, (**c**) HER2-2, and (**d**) HER2-3.

**Figure 2 diagnostics-12-02825-f002:**
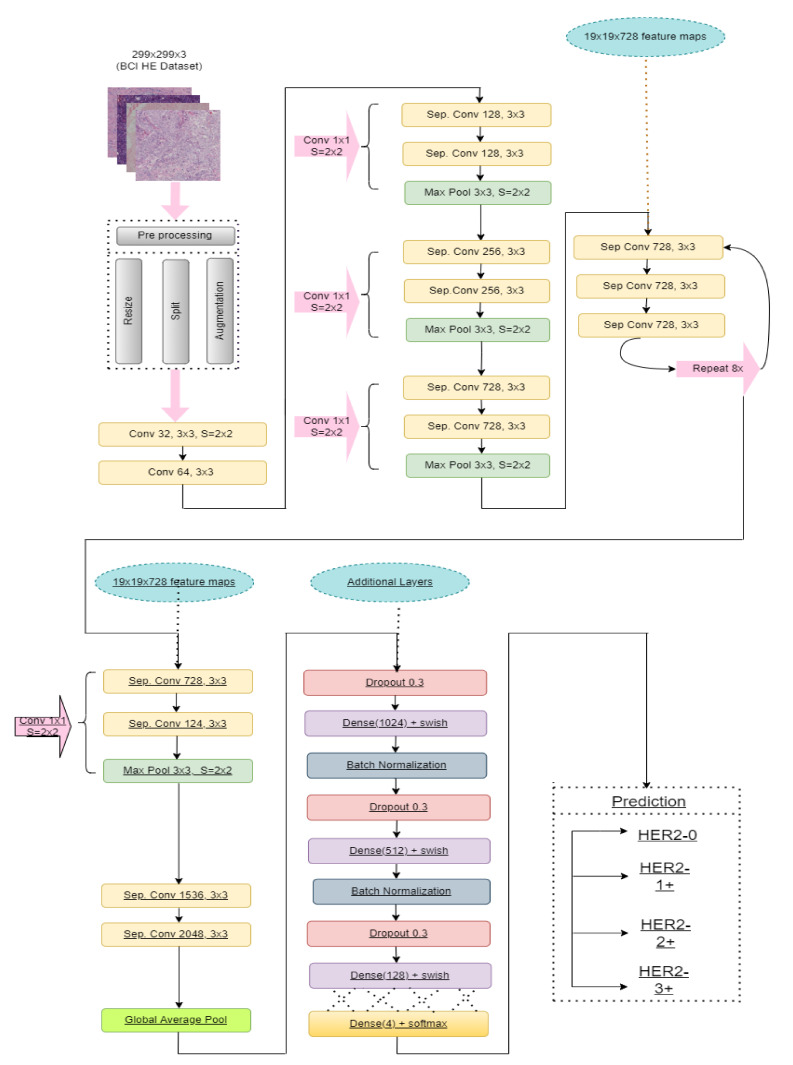
This figure illustrates the proposed model ‘HE-HER2Net’.

**Figure 3 diagnostics-12-02825-f003:**
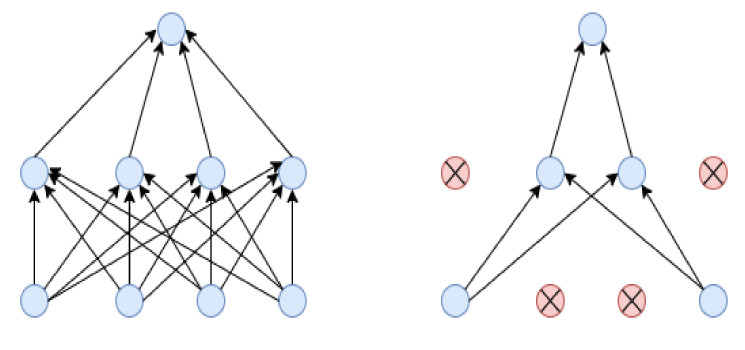
This figure illustrates how the dropout layer works.

**Figure 4 diagnostics-12-02825-f004:**
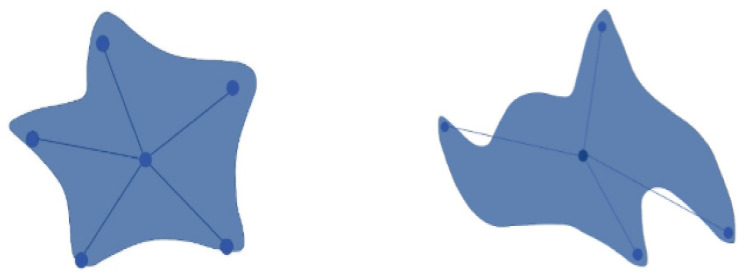
This figure defines the difference between star-convex objects and non-star-convex objects.

**Figure 5 diagnostics-12-02825-f005:**
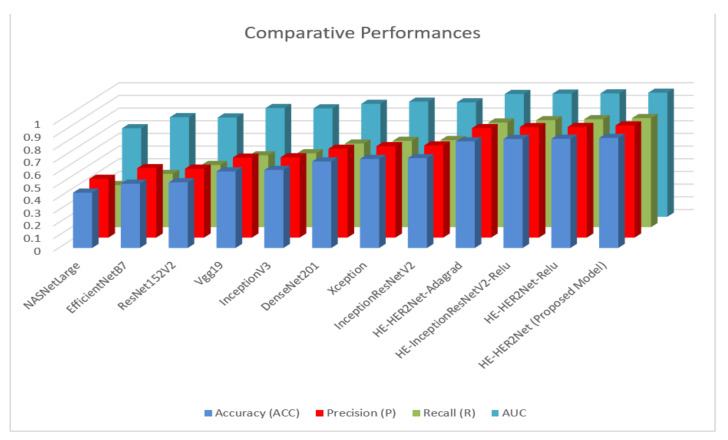
This column chart briefly describes comparative performances among all models. ‘HE-HER2Net’ is the proposed model.

**Figure 6 diagnostics-12-02825-f006:**
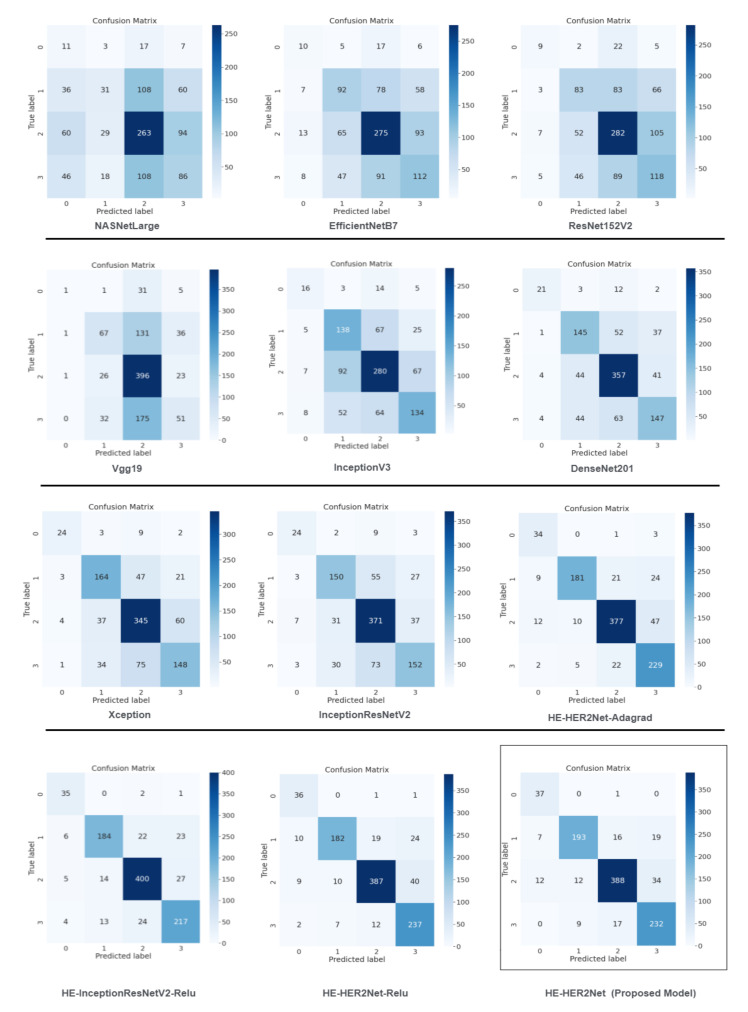
This broad figure shows the performance of the confusion matrix of all the models, including ‘HE-HER2Net’.

**Figure 7 diagnostics-12-02825-f007:**
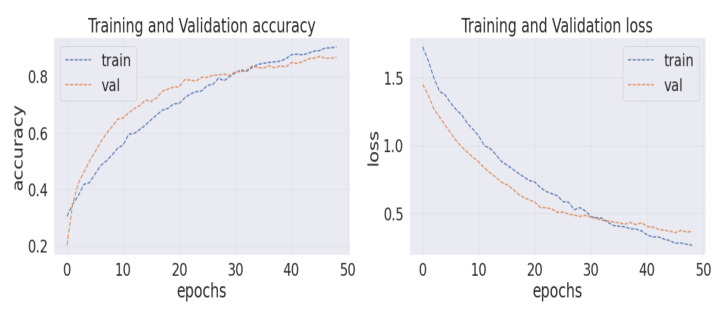
The left corner of the image represents the training and validation accuracy graph, and the right corner shows the training and validation loss graph.

**Figure 8 diagnostics-12-02825-f008:**
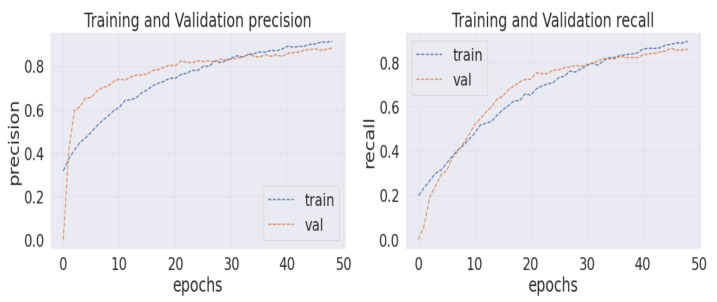
The left corner of the image represents the training and validation precision graph, and the right corner shows the training and validation recall graph.

**Figure 9 diagnostics-12-02825-f009:**
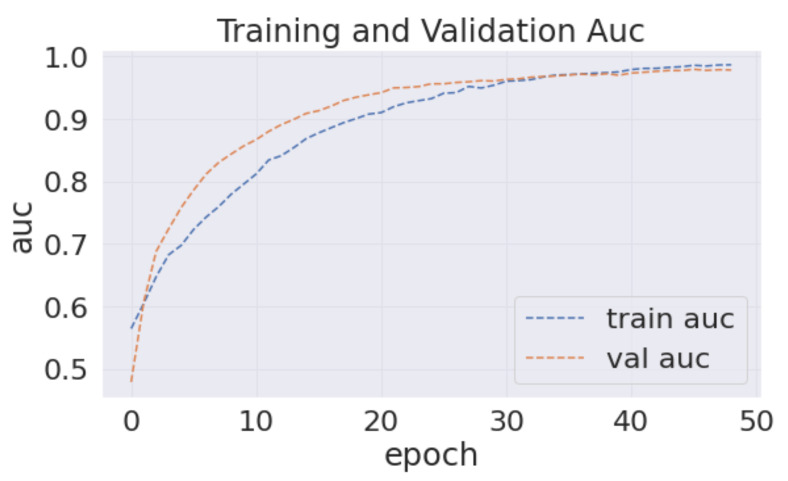
This figure demonstrates the training and validation value of AUC.

**Figure 10 diagnostics-12-02825-f010:**
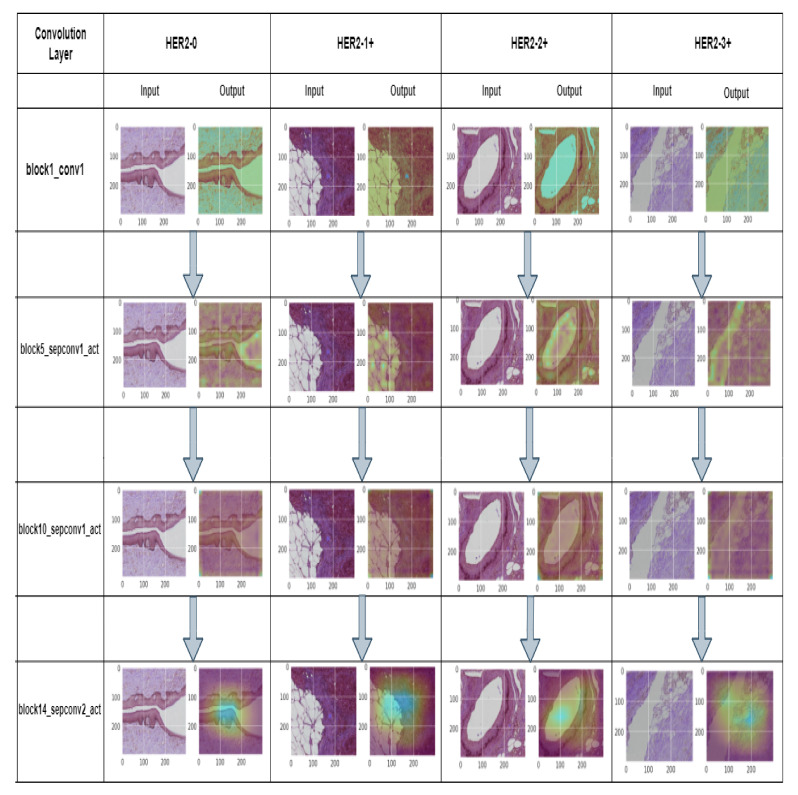
This figure explains our proposed model-‘HE-HER2Net’, for different convolution layers generating heatmaps.

**Figure 11 diagnostics-12-02825-f011:**
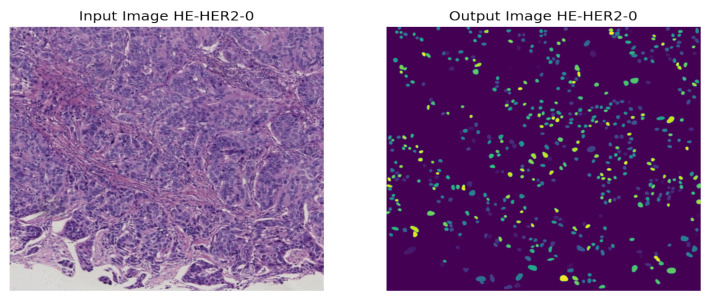
Nuclei segmentation of HER2-0 Breast Cancer.

**Figure 12 diagnostics-12-02825-f012:**
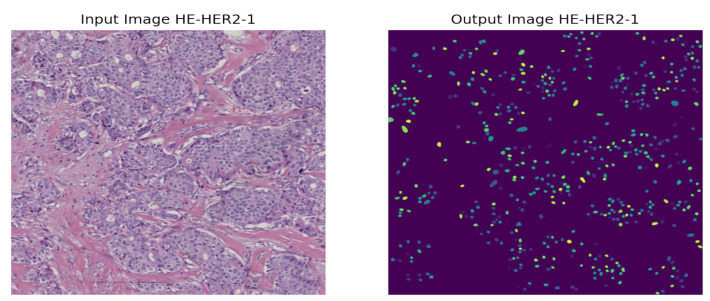
Nuclei segmentation of HER2-1+ Breast Cancer.

**Figure 13 diagnostics-12-02825-f013:**
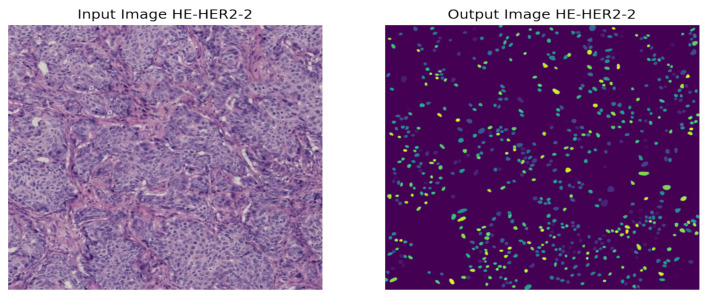
Nuclei segmentation of HER2-2+ Breast Cancer.

**Figure 14 diagnostics-12-02825-f014:**
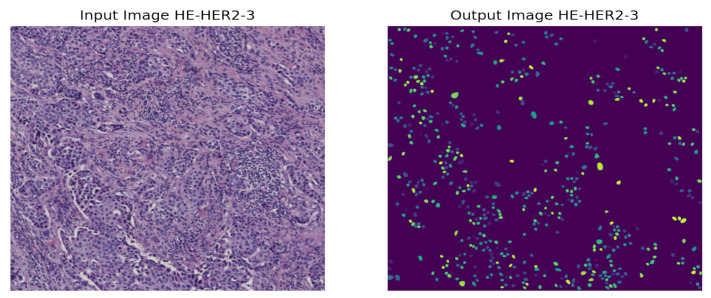
Nuclei segmentation of HER2-3+ Breast Cancer.

**Table 1 diagnostics-12-02825-t001:** This table describes the summary of the environmental setup.

Method	GPU Name	Input Size (pixels)	Batch Size	Optimizer & Learning Rate (lr)	Epoch & Early Stopped	Activation Function	Data Augmentation
Vgg19	Tesla T4	299 * 299	16	Adam, lr: 1 × 10^−5^	50 [ES:6]	ReLU	Not Applied
NASNetLarge	Tesla T4	331 * 331	16	Adam, lr: 1 × 10^−5^	50 [ES:4]	ReLU	Not Applied
EfficientNetB7	Tesla T4	224 * 224	16	Adam, lr: 1 × 10^−5^	50 [ES:7]	ReLU	Not Applied
InceptionV3	Tesla T4	299 * 299	16	Adam, lr: 1 × 10^−5^	50 [ES:11]	ReLU	Not Applied
ResNet152V2	Tesla T4	331 * 331	16	Adam:lr: 1 × 10^−5^	50 [ES:7]	ReLU	Not Applied
InceptionResNetV2	Tesla T4	299 * 299	16	Adam, lr: 1 × 10^−5^	50 [ES:20]	ReLU	Not Applied
DenseNet201	Tesla T4	299 * 299	16	Adam, lr: 1 × 10^−5^	50 [ES:17]	ReLU	Not Applied
Xception	Tesla T4	299 * 299	16	Adam, lr: 1 × 10^−5^	50 [ES:11]	ReLU	Not Applied
HE-InceptionResNetV2-ReLU	Tesla T4	299 * 299	16	Adam, lr: 1 × 10^−5^	80 [ES:41]	ReLU	Applied
HE-HER2Net-ReLU	Tesla T4	299 * 299	16	Adam, lr: 1 × 10^−5^	80 [ES:59]	ReLU	Applied
HE-HER2Net-Adagrad	Tesla T4	299 * 299	16	Adagrad:lr 1 × 10^−3^	80 [ES:30]	swish	Applied
HE-HER2NET (Proposed Model)	Tesla T4	299 * 299	16	Adam, lr: 1 × 10^−5^	80 [ES:49]	swish	Applied

**Table 2 diagnostics-12-02825-t002:** This table describes the output shape and parameters of the additional layers of the proposed model.

Additional Layers	Output Shape	Parameters
global_average_pooling2d_1 (GlobalAveragePooling2D)	(None, 2048)	0
dropout_3 (Dropout)	(None, 2048)	0
dense_4 (Dense)	(None, 1024)	2,098,176
batch_normalization_6 (BatchNormalization)	(None, 1024)	4096
dropout_4 (Dropout	(None, 1024)	0
dense_5 (Dense)	(None, 512)	524,800
batch_normalization_7 (BatchNormalization)	(None, 512)	2048
dropout_5 (Dropout)	(None, 512)	0
dense_6 (Dense)	(None, 128)	65,664
dense_7 (Dense)	(None, 4)	516

**Table 3 diagnostics-12-02825-t003:** This table illuminates the performance evaluation of all the models, such as accuracy (ACC), precision (P), recall (R), and AUC. The bold sentence in the below column of the table represents the proposed model.

CNN_Model	Accuracy	Precision	Recall	AUC
NASNetLarge	0.44	0.46	0.33	0.70
EfficientNetB7	0.51	0.55	0.42	0.79
ResNet152V2	0.52	0.54	0.49	0.78
Vgg19	0.60	0.63	0.57	0.86
InceptionV3	0.61	0.63	0.58	0.85
DenseNet201	0.68	0.70	0.66	0.89
Xception	0.70	0.72	0.68	0.91
InceptionResNetV2	0.71	0.73	0.69	0.90
HE-HER2Net-Adagrad	0.84	0.86	0.82	0.97
HE-InceptionResNetV2-ReLU	0.86	0.87	0.84	0.97
HE-HER2Net-ReLU	0.86	0.87	0.85	0.97
**HE-HER2Net** **(Our Proposed Model)**	**0.87**	**0.88**	**0.86**	**0.98**

## Data Availability

Publicly available datasets were analyzed in this study. This data can be found here: https://bupt-ai-cz.github.io/BCI/.

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
