# Peer review of "Strategies for Enhancing the Multi-Stage Classification Performances of HER2 Breast Cancer from Hematoxylin and Eosin Images"

_diagnostics, 2022, doi:10.3390/diagnostics12112825_

Round 1

Reviewer 1 Report

This manuscript is according to the stated contribution presenting a new improved neural network architecture by introducing “Additional Global Average Pooling Layer, Dropout layers, Batch Normalization layers, Dense layers with a Swish activation function, and Classifier layer with SoftMax activation layer.” They are also explaining the model through Grad-CAM and additionally visualize the nuclei through implementing nuclei segmentation using the StarDist method. The resulting network show better performance than the architectures from literature that they compare it to. So the results of the paper may be worth publishing.

But unfortunately the general presentation in the paper is hard to follow. This reviewer is not a native English speaker so I should not criticize and try to correct the English of the paper. And in general most sentences seem linguistically correct. But I am not sure they always are semantically correct. A clear example is already in the introduction, line 20 on page 1 where it is stated “By 2040 the number of cases is desired to increase by more than 46%” I do not think anyone desires the number of breast cancer cases to increase by 46%. Probably the authors mean “predicted to increase”. In many other places in the paper I find similar somewhat strange statements so that I do not really understand what the authors want to say.

This difficulty in understanding the message of the manuscript may be the reason why I get the feeling that the authors are not fully familiar with the context of their work. For instance they are already in the title of the paper talking about “HER2 breast cancer”. But HER2 is a special protein that can be more or less expressed in breast cancer. And cancer that expresses that protein responds to treatment differently from cancers that do not express this protein, i.e. are HER2 negative.  The methods that they are developing has the goal of determining if a cancer is HER2 positive or negative or uncertain.

Another strange part of the paper is that they in the end of the paper implement a method for segmenting cell nuclei. But there is no discussion about for what purpose you are segmenting the nuclei. Typically that is used as a first processing step if you have some analysis method that needs detected and segmented nuclei for the assessment. But the network they present does not seem to need segmented nuclei.

I strongly suggest someone with an in depth understanding of breast cancer and a really good knowledge of English takes a look at the paper and give some advice on how to improve the structure to make it more understandable.

In addition to those general comments I have some more technical questions and suggestions:

An unclear aspect of the paper is what type of staining they are working with. They state that the dataset contains 4870 pairs of pathological sliced frames stained with H%E and IHC but it is not clear whether they are using both stains or only H&E in their work.

Also all the work is performed on a dataset with around 1200 images for each of stains and the four categories used for the resulting classification. That is a rather small dataset for both training and testing a CNN system.  The authors have spent a lot of work on several levels in optimizing the network for this dataset so it is likely that it is overfitted to the dataset in general even though the results are presented for a held out test set. The results would be much more convincing if the developed network could be tested on some independent data.  

Are all the other networks they are comparing their methods to run on the same dataset? Please be more clear about how the results for the different networks have been obtained.

Another small technical detail is that the results from the performance tests are presented with four decimals. That is more than what is supported by the number of cases that are available. 1200 cases in each group makes four decimal cases a maximum. And some statistical significance tests with confidence intervals would make it more clear how significant the differences between the different methods really are.   

Always when working with microscopy images it is important to specify what optical and digital resolution is being used. It is stated that the images in the database are 1024x124 pixels. But nothing is said about how many microns each pixel are. And in the CNN pipelines the images are 299x299 or 331x331 but nothing is said about how those smaller images are extracted from the 1024x1024 images. By random or deterministic cropping? By subsampling? Or some other method?

Finally, the paper is full of acronyms and abbreviations some of which are explained the first time they are used but not all. Since the paper addresses both medical and AI communities it is likely that many readers are unfamiliar with many of the terms used. I recommend that you introduce a table explaining all the acronyms and their meaning.

Reviewer 2 Report

A transfer learning-based model called ‘HE-HER2Net’ is proposed to diagnose multi-stages of HER2 breast cancer (HER2-0, HER2-1+, HER2-2+, HER2-3+) on H&E (hematoxylin & eosin) image of the BCI dataset. It is more like a combination of existing methods and the novelty is limited. 

 I vote for major revision for the following reasons.  

  1. Separate a 'related work' section to describe references.
  1. List the number of parameters of each layer in your network.
  1. Describe the limitation of this work.
  2. Show some failure examples and explain the reason.
  3. Compare with other state-of-the-art methods.

Reviewer 3 Report

The paper is nicely written and prepared. The topic of the manuscript is of high interest.

The title should be extended. It sounds too general.

line 74: "While using a small dataset for training, a method known as Transfer Learning (TL) can shorten training time and enhance model performance". This sentence is questionable since these "transfer" systems are trained on huge volumes. And then a trained "body" is presented to a broad audience.

Starting with line 89 (up to line 104), the recent works' analysis is given sloppily. It is recommended to dwell on every source in detail.

The paper's description is extremely shallow at the end of the Introduction section.

The article's novelty must be formulated in a more precise way. The current contribution does not represent how unique the offered approach is.

The Conclusion should summarize the obtained results and the whole research. The current Conclusion is not a summary of the fulfilled workflow. The text is significantly similar to the Abstract which is a bad sign.

A separate section for the limitations of the approach should be added.

The Discussion is rather obvious and shallow. There are lots of graphs and tables which are not explained properly. Some in-depth analysis is required here. Try to make conclusions based on the obtained outcomes.

A value of Z is not explained in the expression (8).

There are some bugs in the presentation connected with tables and expressions. Please take another look at the paper's presentation once again.

Formulas (2)-(5): the explanations for the parameters are missing.

Round 2

Reviewer 1 Report

Thank you for answering my questions and clarifying your work. Still there are some remaining issues that should be handled before publication. The text of the new version of the manuscript still contains English sentences that are hard for me to understand properly. Additionally I have some comments, questions and suggestions relating to some of the issues I raised in my first review:

I was asking about the purpose of segmenting the nuclei. You answer that you at numerous places in the paper mention the segmentation. But the only purpose you describe is to visualize some segmented nuclei as green and yellow binary blobs in figures 11 through 14. You state: “Nuclei segmentation from histopathological images is very important to help pathologists to analyze the cells if they are benign or malignant. Generally, cancer cell nuclei are more extensive and darker compared to normal cells because of containing comprehensive DNA. Thus, nuclei segmentation is an essential task for researchers in digital pathology” How does the visualizations you show help the pathologists determine if the nuclei are more extensive and darker? As far as I can see you do not give any explanation about if the different colors has any significance. Also far from all nuclei in the H&E image are shown in the binary result image. Do you have any results verifying that the method shows relevant images and not a more or less random subset? You are claiming that such visualization is of value to the pathologists. Do you have any support for that? In my experience pathologists prefer to look at the H&E images and at any meaningful quantitative data that may be extracted describing the classification result or similar. The heat map visualizations may for instance be of value in explaining the basis of the results from the CNN. But what would the value of the binary colorful blobs be? They could be used as input of a classifier that extracts features from the nuclei and uses that as a basis for a classification. That is why I stated that segmentation is typically used as a first processing step. But it seems you claim the segmentation is of importance through this visualization. Please clarify.

As a response to my statement that the number of cases you are working with does not motivate four decimal places in the accuracy data you claim that you get the four decimals from the Keras and the tensorflow library. But the fact that a specific program delivers a certain number of digits is no proof that all those digits are valid. It is the responsibility of the researcher to evaluate what accuracy is meaningful. And with a few hundred cases in each category it should probably be two digits. And differences in the second digit only is likely not statistically significant. You can not claim that “our proposed model's results demonstrated significantly better in all evaluation metrics” without performing proper significance tests. “significantly better” has a defined scientific meaning based on statistics.

I still think that the limited size of the available dataset make the validity of the results questionable. But as you say in the reply, it is not easy to find an independent dataset to use for validating the results.

In response to my question about how you handled the different image sizes for the different networks you have now clarified that you resized the images thus loosing resolution. Still you do not give the important information about what the original resolution is. How big is a pixel in the 1024x1024 images in the database?

Thank you for adding the table explaining all the acronyms.
